# The *Phaseolus vulgaris* Receptor-Like Kinase PvFER1 and the Small Peptides PvRALF1 and PvRALF6 Regulate Nodule Number as a Function of Nitrate Availability

**DOI:** 10.3390/ijms24065230

**Published:** 2023-03-09

**Authors:** Jorge Solís-Miranda, Marco A. Juárez-Verdayes, Noreide Nava, Paul Rosas, Alfonso Leija-Salas, Luis Cárdenas, Carmen Quinto

**Affiliations:** 1Departamento de Biología Molecular de Plantas, Instituto de Biotecnología, Universidad Nacional Autónoma de México, Avenida Universidad 2001, Colonia Chamilpa, Cuernavaca, Morelos 62210, Mexico; 2Departamento de Docencia, Universidad Autónoma Agraria Antonio Narro, Saltillo, Coahuila 25315, Mexico; 3Centro de Ciencias Genómicas, Universidad Nacional Autónoma de México, Avenida Universidad 2001, Colonia Chamilpa, Cuernavaca, Morelos 62210, Mexico

**Keywords:** nodulation, autoregulation of nodulation, nitrate-mediated regulation of nodulation, FERONIA, RALF

## Abstract

Legumes associate with Gram-negative soil bacteria called *rhizobia*, resulting in the formation of a nitrogen-fixing organ, the nodule. Nodules are an important sink for photosynthates for legumes, so these plants have developed a systemic regulation mechanism that controls their optimal number of nodules, the so-called autoregulation of nodulation (AON) pathway, to balance energy costs with the benefits of nitrogen fixation. In addition, soil nitrate inhibits nodulation in a dose-dependent manner, through systemic and local mechanisms. The CLE family of peptides and their receptors are key to tightly controlling these inhibitory responses. In the present study, a functional analysis revealed that PvFER1, PvRALF1, and PvRALF6 act as positive regulators of the nodule number in growth medium containing 0 mM of nitrate but as negative regulators in medium with 2 and 5 mM of nitrate. Furthermore, the effect on nodule number was found to be consistent with changes in the expression levels of genes associated with the AON pathway and with the nitrate-mediated regulation of nodulation (NRN). Collectively, these data suggest that PvFER1, PvRALF1, and PvRALF6 regulate the optimal number of nodules as a function of nitrate availability.

## 1. Introduction

Nitrogen-fixing nodules are specialized organs that contain bacteria generically called rhizobia, which colonize the roots of legumes in a symbiotic interaction. Inside this organ, atmospheric dinitrogen (N_2_) is converted into ammonium (NH_4_^+^) through the activity of the nitrogenase complex present in differentiated bacteroids, providing plant hosts with this important source of bioavailable nitrogen [1]. Legumes attract compatible partners by exuding flavonoids into the rhizosphere, which are specifically detected by a bacterial membrane-associated protein, NodD [2]. After this perception, the rhizobia express the nodulation (*nod*) genes, resulting in the biosynthesis and secretion of the Nod factors [3]. Nod factors are in turn specifically perceived by a receptor complex located in the root hair tip membrane, which in *Lotus japonicus* L. includes NOD FACTOR RECEPTOR 1 (LjNFR1), LjNFR5, and SYMBIOTIC RECEPTOR-LIKE KINASE (LjSYMRK) [4,5]. Subsequently, several genes are expressed in the legume, including *NODULE INCEPTION* (*NIN*), which is a master regulator of nodulation necessary for infection, organogenesis, control of the nodule number, metabolism, and senescence [6,7,8].

Nodule organogenesis and nitrogen fixation require high levels of energy. To optimize the cost of energy with the benefits of nitrate uptake, legumes have developed local and systemic regulatory mechanisms to control the optimal number of nodules formed [9]. In *Glycine max* (L.) Merr., *L. japonicus*, and *Medicago truncatula* Gaertn., a systemic regulatory mechanism called autoregulation of nodulation (AON) controls the optimal number of nodules. During this process, RHIZOBIA-INDUCED CLAVATA3/ENDOSPERM-SURROUNDING REGION (CLE) (GmRIC1) and GmRIC2 (which are respectively known as LjCLE-R1 and LjCLE-R2 in *L. japonicus* and MtCLE12 and MtCLE13 in *M. truncatula*) are biosynthesized in the roots of these legumes, induced by the rhizobia [10,11]. GmRIC1 and GmRIC2 are transported to the aerial tissues and perceived by a leucine-rich repeat (LRR)-RLK receptor NODULE AUTOREGULATION RECEPTOR KINASE/SUPER NUMERIC NODULES/HYPERNODULATION AND ABERRANT ROOT (GmNARK/MtSUNN/LjHAR) [12,13,14]. The perception of these peptides leads to a reduction in the level of a mobile microRNA, miR2111, which promotes the accumulation of the mRNA of *TOO MUCH LOVE* (*TML*) in the roots, encoding an F-box protein that negatively regulates nodulation [15,16]. NIN is known to bind directly to the promoter sequences of the genes encoding LjCLE-RS1 and LjCLE-RS2 in *L. japonicus* or *MtCLE13* in *M. truncatula*, activating their expression [6,8].

Nitrogen (N) is essential for plant development, and the presence of nitrate in the soil affects nodulation in a concentration-dependent manner through both local and systemic mechanisms [17,18]. In *G. max*, the production of the NITRATE-INDUCED CLE1 (GmNIC1) peptide is induced in roots in the presence of nitrate and is perceived by GmNARK in the root, inhibiting nodulation through an unknown local mechanism [10]. In *M. truncatula*, nitrate was reported to inhibit rhizobia-mediated *NIN* transcription through the action of NIN-like proteins (NLPs). At high nitrate levels, *NIN* expression is reduced and NLP proteins accumulate in the nucleus, bind to the few remaining NIN proteins, and suppress the expression of the NIN-activated genes, locally inhibiting nodule formation [19]. NLP1 in *M. truncatula* and its homolog NITRATE UNRESPONSIVE SYMBIOSIS1 (NRSYM1) in *L. japonicus* accumulate in the nucleus in response to high levels of nitrate and activate the expression of CLE peptides (CLE35 in *M. truncatula* and CLE-RS2 in *L. japonicus*) by directly binding to their promoter sequences, mediating the inhibition of nodulation through the AON pathway [20,21,22].

In addition to CLEs, other peptides are required for the legume–rhizobia symbiosis. Members of the C-terminally encoded peptide (CEP) family are important for the systemic induction of nodulation under low-nitrate conditions [23]. Nodule-specific cysteine-rich peptides (NCRs), exclusively present in legumes of the inverted repeat–lacking clade (IRLC), are essential for the differentiation of bacteroids [24]. Rapid alkalinization factors (RALFs) are another family of peptides involved in nodulation; for example, in *M. truncatula*, the overexpression of *MtRALF1* caused an increase in aborted infections, a reduction in the number of nodules, and aberrant nodulation [25]. The mechanism by which mtRALF1 inhibits nodulation and whether this mechanism is conserved in other legumes are yet to be determined. Using a phosphoproteomic assay in *Arabidopsis thaliana* (L.) Heynh, [26] showed that FERONIA (FER), an RLK of the CrRLK1L (*Catharanthus roseus* (L.) G. Don RLK1L) subfamily, is a RALF1 receptor [26]. Furthermore, in a recent report, FER was shown to be required for the perception and function of more than half of the RALFs in *A. thaliana* [27], suggesting the involvement of FER in the regulation of nodulation through its perception of MtRALF1 in *M. truncatula*.

The FER–RALF module is known to regulate plant growth during biotic and abiotic stress conditions, including metal ion, salinity, drought, and mechanical stresses [28,29,30]. Interestingly, the plant pathogenic fungi *Golovinomyces orontii* and *Magnaporthe oryzae*, as well as the plant-parasitic nematode *Meloidogyne incognita*, produce RALF-like peptides (F-RALFs and MiRALFs, respectively) that bind to FER during their infection of *A. thaliana*, increasing their pathogenicity [31,32,33]. Recently, we reported that different *CrRLK1Ls*, including the closest *FER* homolog, are expressed in the nodules of four legumes [34], suggesting a possible role for the FER–RALF module during this symbiotic process.

Herein, we found that *PvFER1*, *PvRALF1*, and *PvRALF6* are expressed in *Phaseolus vulgaris* L. nodules, which respond differentially to *Rhizobium tropici* inoculation under low- versus high-nitrate conditions. We also investigated the role of these three genes in the regulation of the optimal number of nodules as a function of a suitable or low-nitrate concentrations in the growth medium. Through RNA interference (RNAi) silencing and overexpression of *PvFER1*, *PvRALF1*, or *PvRALF6*, these genes were found to positively or negatively regulate the expression of downstream genes involved in the nitrate-mediated regulation of nodulation (NRN) and AON pathways under nothing and high-N conditions, respectively.

## 2. Results

### 2.1. PvFER1 Is a P. vulgaris RLK Expressed in Nodules

Recently, we identified 33 *CrRLK1L* genes in the *P. vulgaris* genome, eight of which are expressed in roots [34]. A comparison of *CrRLK1L*s expression in four legume species (*P. vulgaris*, *L. japonicus*, *G. max*, and *M. truncatula*) indicated that a gene homologous to the *A. thaliana* gene *FER* was expressed in all of their nodules [34]. Using a qPCR analysis, we also demonstrated that *PvFER1* (Phvul.008G08100) was expressed in *P. vulgaris* roots inoculated with *R. tropici*, throughout various stages of nodule development [34]. An *in-silico* analysis of the primary structure of PvFER1 showed that it possesses a transmembrane domain, a characteristic extracellular malectin-like domain, and a cytoplasmic kinase domain (Appendix A), as is reported for FER in *A. thaliana*. These data prompted us to evaluate the role of *PvFER1* in nodulation in *P. vulgaris*.

The spatial expression patterns of *PvFER1* were investigated in transgenic *P. vulgaris* roots and nodules by analyzing the expression of a *GUS* reporter driven by the *PvFER1* promoter (*proPvFER1*::GUS). GUS activity was analyzed in the hairy roots and nodules of the composite plants grown in 0 mM nitrate. *PvFER1* promoter activity was observed in the apices and the central cylinder of the main and lateral roots (Appendix A). In inoculated roots, the promoter activity was observed in the primordia of nodules during all stages evaluated (Figure 1A–C), as well as in the vascular bundles of the mature nodules (Figure 1D). Collectively, these data suggest that PvFER1 is a RLK expressed in the meristematic zones and vasculature of *P. vulgaris* roots and nodules.

### 2.2. PvRALF1 and PvRALF6 Are Expressed in Nodules and Are Cysteine-Rich Peptides That Interact with PvFER1

It is well known that RALF peptides are ligands for CrRLK1L receptors, and there have also been reports of interaction of RALFs of a given plant species with a CrRLK1L of a different one [27,35,36,37,38]. To assess whether RALF contributes to the regulation of bean nodule number, we first examined the *RALF* gene family in *P. vulgaris*, revealing nine members that encode peptides with high similarity to RALF1 from *A. thaliana*. All nine PvRALFs showed conserved RRXL and YISY signatures, as well as a conserved secretion signal and four cysteine residues (Appendix A). A phylogenetic analysis of RALF amino acid sequences from *A. thaliana*, *P. vulgaris*, *M. truncatula*, and *G. max* and the RALF from the fungal plant pathogen *Fusarium proliferatum* grouped the nine PvRALFs into three of the four major clades formed (Appendix A); clade 1 contains four PvRALFs, clade 2 contains two PvRALFs, clade 3 contains three PvRALFs, while clade 4 contains no PvRALFs. These data are consistent with previous analyses of RALF peptides in 52 different plant species [39].

The expression profiles of the nine *PvRALF* genes were obtained from the *P. vulgaris* Gene Expression Atlas [40]. Five of the genes, *PvRALF2*, *PvRALF4*, *PvRALF5*, *PvRALF6*, and *PvRALF8*, show little or no expression, while *PvRALF1*, *PvRALF3*, *PvRALF7*, and *PvRALF9* were expressed at high levels in all tissues tested (Appendix A). Among the four most expressed genes, *PvRALF1* (Phvul.007G197000), the closest homolog of *A. thaliana* RALF1, was the most abundant in inoculated roots and nodules. Interestingly, *PvRALF6* (Phvul.001G266400), the closest homolog of *MtRALF1*, showed the highest expression levels in nodules at 5 dpi. RT-qPCR analysis of *P. vulgaris* roots inoculated with *R. tropici* or a mock solution and grown in 0 mM nitrate confirmed that the *PvRALF1* and *PvRALF6* transcripts accumulated in differing patterns following the bacterial inoculation (Figure 2A,B); the *PvRALF1* transcript abundance increased in the inoculated roots at 3 and 7 dpi and decreased at 5 and 14 dpi (Figure 2A), while *PvRALF6* only showed an increase at 14 dpi (Figure 2B) compared with the mock-inoculated roots. These results suggest that *RALF* genes could fine-tune the different stages of nodulation.

The promoter activity of each of these two genes was analyzed by evaluating the spatial expression patterns of pro*PvRALF1*::GUS and pro*PvRALF6*::GUS using a histochemical analysis of GUS activity in *P. vulgaris* hairy roots. Similar to the results observed in the analysis of pro*PvFER1*::GUS, the GUS activity of pro*PvRALF1*::GUS and pro*PvRALF6*::GUS was observed in the root apices, the central cylinder of the roots (Appendix A), and in the nodules during the evaluated times (Figure 1E–L). These observations, together with previous reports in *A. thaliana* posing FER as the main receptor for most RALF peptides [27], strongly suggest that PvRALF1 and/or PvRALF6 could interact with PvFER1 in roots and nodules. This prompted us to confirm whether PvRALF1 and/or PvRALF6 interact(s) with PvFER1 using the split ubiquitin system. The observed growth of yeast indicated that Cub-PvFER1ΔK (a truncated version of PvFER1 that lacks the kinase domain to avoid undesirable effects in yeast) was able to restore spliced ubiquitinase function when NubWT, PvRALF1-Nub, or PvRALF6-Nub was present, but not when a mutated version of the Nub protein was used (NubΔG) (Figure 2C). These results confirm that PvRALF1 and PvRALF6 interact with PvFER1, strongly suggesting that these three proteins together could participate in nodulation in *P. vulgaris*.

### 2.3. PvFER1, PvRALF1, and PvRALF6 Contribute to Determining the Number of Nodules Produced by P. vulgaris Roots

To assess the role of *PvFER1* in nodulation, the effects of the RNAi-mediated downregulation and overexpression of this gene in *P. vulgaris* hairy roots grown under 0 mM of nitrate were investigated. The expression levels of this gene in the *PvFER1-*silenced (*PvFER1i*) and *PvFER1-*overexpressing (*PvFER1ox*) lines were first confirmed using a qPCR analysis, revealing an approximately 50% reduction in the *PvFER1* transcripts in the *PvFER1i* roots and a five-fold increase in the *PvFER1ox* roots (Appendix A). Importantly, no differences were observed in the length of the *PvFER1i* or *PvFER1ox* roots compared with the controls (Appendix A). An analysis of the number of nodules in these lines revealed significant differences in the total number of nodules generated by these composite plants (Figure 3A,B); *PvFER1i* produced fewer nodules than the control roots at 21 and 31 dpi (Figure 3A), while the opposite was true for *PvFER1ox* (Figure 3B).

Similar to *PvFER1*, composite bean plants with *PvRALF1* RNAi (*PvRALF1i*), *PvRALF6* RNAi (*PvRALF6i*), *PvRALF1*-overexpressing (*PvRALF1ox*), or *PvRALF6*-overexpressing (*PvRALF6ox*) roots and their respective controls were inoculated with *R. tropici*-GUS and grown in the absence of nitrate. An RT-qPCR analysis showed approximately 70% and 80% reductions in the transcript accumulation levels of the *PvRALF1i* and *PvRALF6i* roots, respectively, and 35- and 200-fold increases in the *PvRALF1ox* and *PvRALF6ox* roots, respectively (Appendix A). Similar to *PvFER1*, the *PvRALF1/PvRALF6-*silenced and -overexpressing roots showed no difference in root length to the controls (Appendix A). *PvRALF6*-silenced lines inoculated with *R. tropici* showed a reduction in the number of nodules at 21 and 31 dpi compared with the control, while *PvRALF1i* showed no significant difference (Figure 3C,E). In contrast, *PvRALF1ox* displayed an increased number of nodules at 21 and 31 dpi compared to the control and transgenic roots overexpressing *PvRALF6* which showed no changes (Figure 3D,F).

To assess the additional effects of these different levels of *PvFER1*, *PvRALF1i*, and *PvRALF6i* transcript accumulation on nodule organogenesis and nitrogenase activity, we analyzed the diameters and acetylene reduction of the nodules produced by these different transgenic roots. Only small differences in nodule diameters were found in both conditions compared with the control nodules (Appendix A). Furthermore, non-significant differences in acetylene reduction were observed in the nodules of roots overexpressing *PvRALF1* and *PvRALF6* or silenced in *PvFER1* (Appendix A).

### 2.4. PvFER1, PvRALF1, and PvRALF6 Regulate the Expression of AON-Related Genes

To estimate the involvement of *PvFER*, *PvRALF1* and *PvRALF6* in nodulation signaling pathways, we examined the effect of silencing and overexpressing of these genes on the expression of key genes related to nodule organogenesis (*nodule inception* (*PvNIN*) and *leghemoglobin* (*PvLEG*)), nodule metabolism (*sucrose synthase* (*PvSS*) and *glutamine synthetase* (*PvGS*)), and AON (*PvRIC2* and *too much love* (*PvTML*)). When comparing the expression of genes involved in organogenesis and metabolism of nodules in the *PvFER1i* and *PvFER1ox* roots with their controls, no differences were observed in the expression of the *PvNIN* and *PvSS* genes; however, *PvLEG* and *PvGS* were increased and decreased in *FER1i* and *PvFER1ox* roots, respectively (Appendix A). No significant differences in the expression levels of *PvNIN*, *PvLEG*, *PvGS*, or *PvSS* were detected in the *PvRALF1i*/*PvRALF6i* and *PvRALF1ox/PvRALF6ox* roots (Appendix A).

The expression of the AON marker gene *PvRIC2* was increased in *PvFER1i* and reduced in the *PvFER1ox* roots compared with the controls (Figure 4A,B), while the accumulation of *PvTML* transcripts was similar between the *PvFER1i* and *PvFER1ox* roots and the controls (Figure 4A,B). *PvRIC2* expression was increased in the *PvRALF1i* roots and reduced in the *PvRALF1ox* roots compared with the controls, whereas *PvRIC2* expression was higher in the *PvRALF6ox* roots (Figure 4A,B). On the other hand, no change in *PvTML* expression was detected in the *PvRALF1i*/*PvRALF6i* and *PvRALF1ox/PvRALF6ox* lines (Figure 4A,B). Collectively, these data suggest that *PvFER1*, *PvRALF1*, and *PvRALF6* participate in the regulation of the optimal number of nodules in bean roots by somehow modulating the expression of *PvRIC2*, while no clear effect was observed on the nodule organogenesis or metabolism genes.

### 2.5. PvFER1, PvRALF1, and PvRALF6 Respond to Symbiosis under High-Nitrate Conditions

Previously, it was reported that FER and RALF1 regulate *A. thaliana* growth under high-C/N relation conditions and nitrate starvation [41,42]. This led us to evaluate the effect of nitrate on the accumulation of *PvFER1*, *PvRALF1*, and *PvRALF6* transcript levels in *P. vulgaris.* A qPCR analysis of roots inoculated with *R. tropici* and grown at 5 mM of nitrate revealed a greater accumulation of *PvRALF1* and *PvRALF6* transcripts at 5, 7, and 14 dpi (Figure 5B,C) compared with plants grown under 0 mM nitrate (Figure 2A,B). In contrast, *PvFER1* showed reduced levels of transcript accumulation under high-nitrate conditions (5 mM) at 3, 5, and 14 dpi (Figure 5A). These observations indicate that the expression of *PvFER1*, *PvRALF1*, and *PvRALF6* are regulated in response to both nitrate and rhizobia, suggesting a role for these genes in regulating nodulation under high-nitrate conditions.

### 2.6. PvFER1, PvRALF1, and PvRALF6 Regulate the Expression of NRN-Related Genes

To investigate the potential involvement of the three analyzed genes in the NRN pathway, the effects of silencing and overexpressing *PvFER1*, *PvRALF1*, and *PvRALF6* on the expression of the NRN marker genes *PvCLE35*, *NODULE-INDUCED CLE1* (*PvNIC1*), and *PvNARK* under 0 mM nitrate were examined. All genes tested showed increased expression levels in transgenic roots in which either *PvFER1*, *PvRALF1*, or *PvRALF6* were silenced, whereas the overexpression of *PvFER1* and *PvRALF1* reduced the expression of *PvNIC1;* no significant changes were detected in the transcript levels of *PvCLE35* or *PvNARK* in the *PvFER1-*, *PvRALF1-*, or *PvRALF6-*overexpressing roots (Figure 4A, B). These results suggest that *PvFER1*, *PvRALF1*, and *PvRALF6* also contribute to the regulation of the expression of the genes in the NRN pathway.

### 2.7. PvFER1, PvRALF1, and PvRALF6 Regulate the Number of Nodules Produced by P. vulgaris under High-Nitrate Conditions

To examine a probable role by which *PvFER1*, *PvRALF1*, and *PvRALF6* regulate nodule number as a function of nitrogen availability, transgenic roots in which these genes were either silenced or overexpressed were challenged by growth under high-nitrate conditions. The *PvFER1i*, *PvRALF1i*, and *PvRALF6i* lines developed more nodules than the control roots under 5 mM of nitrate at 21 and 31 dpi (Figure 6A–C), whereas transgenic roots that overexpressed *PvRALF1* or *PvRALF6*, but not *PvFER1*, showed fewer nodules than the control roots under these same conditions (Figure 6D–F). Similar results were observed with 2 mM of nitrate in both the silenced and overexpressed roots; however, the inhibition of the nodule number was reduced (Appendix A). No significant differences in nodule diameters were observed in any of these transgenic lines grown under 2 or 5 mM of nitrate (Appendix A).

The expression of the nodule organogenesis marker genes (*PvNIN* and *PvLEG*) remained virtually unchanged in the roots in which *PvFER1*, *PvRALF1*, or *PvRALF6* was silenced or overexpressed when grown under high-nitrate conditions (Appendix A). In *PvRALF1ox* roots, an increase in the expression of *PvGS* and *PvSS* was observed; however, their expression was not affected in the *PvFER1ox* or *PvRALF6ox* lines, nor in *PvFER1i*, *PvRALF1i*, or *PvRALF6i* (Appendix A). The expression of the AON marker gene *PvRIC2* was elevated in the *PvFER1-* and *PvRALF1-*silenced roots (Figure 7A), while the *PvTML* transcripts were reduced in the *PvFER1-*, *PvRALF1-*, and *PvRALF6-*silenced roots under this nodulation-inhibitory condition (Figure 7A). When *PvRALF1* was overexpressed, an increase in the accumulation of *PvRIC2* and *PvTML* transcripts was detected (Figure 7B). *PvNARK* transcript levels were reduced in the silenced roots of any of the three genes; however, no changes were detected in the overexpressing roots (Figure 7A,B). The transcript levels of the NRN marker gene *PvNIC1* showed a reduction in the *PvRALF1-*silenced roots and an increase in the *PvRALF1*-overexpressed roots (Figure 7A,B). Finally, the expression of *PvCLE35* did not change in the silenced roots of any of the three genes examined, but was increased in the roots that overexpressed *PvFER1* or *PvRALF1* (Figure 7A,B). These data strongly suggest that PvFER1, PvRALF1, and PvRALF6 participate in the control of the number of nodules by regulating the expression of genes of the AON and NRN pathways at high-nitrate concentrations (5 mM).

## 3. Discussion

The RALF family of peptides and FER, which is a member of the CrRLK1L receptor subfamily, are highly conserved and have, in the last decade, been shown to be involved in a plethora of processes. The interaction between RALF1 and FER was first described in the inhibition of root elongation in *A. thaliana* [43]. Since then, there has been a rapid increase in the number of processes in which these ligand–receptor complexes have been reported to be involved, as recent review articles show [28,29,30]. Despite this research interest, the role of the FER–RALF ligand–receptor complex during symbiotic associations, such as legume–rhizobia symbioses, has been poorly studied. Here, we report that *PvFER1*, *PvRALF1*, and *PvRALF6* participate in the regulation of nodule number in *P. vulgaris* as a function of nitrate availability.

It is well known that RALF1 sensing by FER promotes the inhibition of root elongation and root hair tip growth [26,43,44,45,46]. This is important for nodulation because root hairs constitute the main site at which rhizobia enter legume roots during symbiosis. In the present work, we found that *PvFER1*, *PvRALF1*, and *PvRALF6* are expressed in *P. vulgaris* roots, and the activity of their promoters was also detected in the central cylinder and root apices (Appendix A), similar to what was reported for their homologs in *A. thaliana* and rice (*Oryza sativa* L.) [44,47]. These observations suggest that these genes may have a role in root development in *P. vulgaris*. Haruta et al. (2014) showed that the exogenous addition of synthetic RALF inhibits root expansion in *A. thaliana*, a process that depends on the function of FER; however, in transgenic *P. vulgaris* roots with either silenced or overexpressed *PvFER1*, *PvRALF1*, or *PvRALF6* genes, we observed no differences in root growth compared with the control transgenic roots (Appendix A). Considering that there are 17 *CrRLK1Ls* and 33 *RALFs* in *A. thaliana* [26,27,39,48], and 37 *CrRLK1Ls* and nine *PvRALFs* in the *P. vulgaris* genome [34,39], the apparent lack of phenotype in *P. vulgaris* roots lacking or overexpressing *PvFER1*, *PvRALF1*, or *PvRALF6* transcripts is likely due to the overlapping functions of the remaining *CrRLK1L*s and *RALF* genes. An alternative explanation could be that the remaining transcript levels in the silenced roots (Appendix A) may be sufficient to mediate normal root growth, although no phenotype was observed in the hairy root system control used here. An analysis of *P. vulgaris* plants lacking or overexpressing multiple *RALF*s and *CrRLK1L*s could help answer this question and may also reveal novel functions for the other *RALF* and *CrRLK1L* genes in *P. vulgaris*.

In *M. truncatula*, *MtRALF1* expression was reported to be induced during nodulation, and its overexpression impairs the symbiosis between this legume and the bacterium *Sinorhizobium meliloti* [25]. In line with this, during a comparative analysis of the expression of *CrRLK1L*s in four legumes, a cluster of *CrRLK1L* genes expressed in nodules was found, with the expression of the closest homolog of *A. thaliana FER* detected in the nodules of all legumes tested [34]. An analysis of the expression and promoter activity of *PvFER1*, *PvRALF1*, and *PvRALF6* in inoculated *P. vulgaris* roots revealed that they are induced in nodules during various stages of their organogenesis (Figure 1 and Figure 2A,B). Several reports support the interaction between the RALF peptides and CrRLK1Ls; nearly half of the *A. thaliana* RALFs (16 of 33) can be perceived by FER, leading to the inhibition of root elongation [27]. Along the same line, RALF peptides from different plant species and those from fungi and *M. incognita* can interact with FER from *A. thaliana* [31,32,33]. Through protein–protein interaction analyses, we found that PvRALF1 and PvRALF6 interact with PvFER1 in *P. vulgaris* (Figure 2C), suggesting a role of the PvRALF1/6 perception by PvFER1 at different stages of nodulation in *P. vulgaris*.

Reverse genetic analysis revealed a role for *PvFER1*, *PvRALF1*, and *PvRALF6* in regulating the number of nodules under low- and high-nitrate conditions (Figure 3 and Figure 6); however, little or no effect on nodule development and function was observed (Appendix A). In *M. truncatula*, it was previously reported that the overexpression of *MtRALF1* impairs the symbiosis between this legume and *S. meliloti*, affecting not only the number of nodules, as we observed in *P. vulgaris*, but also impacting their function and decreasing their bacteroid content [25]. The differences observed in nodulation between these two legumes could be due to the different types of nodules that each of them forms; namely, indeterminate nodules in *M. truncatula* and determinate nodules in *P. vulgaris*. It is well known that there are significant differences at the genetic, morphological, and physiological levels between these two types of nodules [49,50]. A further exploration of the role of the *FER* and *RALF* genes in other legumes could answer the questions of whether there is a conserved role of these two genes in nodulation and whether they have additional functions in the development of indeterminate nodules.

Many genes have been reported to be associated with the regulation of nodule number, participating in both the local and systemic signaling pathways involved in the induction or repression of nodulation. The loss of function of the *CLEs*, *NARK*, or *TML* genes in several legumes induced an increase in the number of nodules [10,11,12,13,14], contrary to what was observed with the silencing of *PvFER1*, *PvRALF1*, and *PvRALF6* but similar to their overexpression under 0 mM of nitrate (Figure 3 and Figure 6). Several authors have reported that the RALF1–FER complex regulates the transcription, splicing, and translation of genes associated with root development, root hair growth, and the responses to biotic and abiotic stresses [51,52,53,54]. Here, we describe that *PvFER1*, *PvRALF1*, and *PvRALF6* are involved in regulating the expression of *NARK*, *TML*, and certain *CLE*s (*RIC2*, *NIC1* and *CLE35*) under 0 and 5 mM of nitrate (Figure 4 and Figure 7). In agreement with this, various authors [54,55,56] reported that among the many differentially expressed genes in *A. thaliana fer1* and *fer4* mutants, or overexpressing *RALF23*, numerous genes encoding PvNARK-like LRR receptors were differentially expressed (Appendix A). Furthermore, three of the seven *A. thaliana TML* genes were upregulated in *fer1*, one was upregulated and one downregulated in *fer4*, and none were differentially expressed in *RALF23*ox when compared with the wild-type control. Interestingly, none of the seven *CLE* genes in *A. thaliana* showed changes in expression in any of the lines tested (Appendix A) [54,55,56]. These data suggest that the transcriptional regulation of the *NARK*s and *TML* by PvRALF1, PvRALF6, and PvFER1 observed in the present work is conserved in *A. thaliana*, whereas the regulation of *CLE* gene expression is likely associated with nodulation in legumes. A more detailed analysis of the transcriptional regulation of the AON and NRN pathway genes by FER and RALF in other legumes will confirm whether this mechanism is conserved during the nodulation process. Furthermore, a global assessment of the effect of changes in *RALF* and *FER* gene expression in *P. vulgaris* and other legumes, determined using an RNA-seq analysis, will lead to a deeper understanding of the role of these genes in nodulation.

FER is known to participate in the regulation of plant growth under low-nitrate conditions by interacting with the TOR/RAPTOR complex and with the E3 ubiquitin ligase ARABIDOPSIS TOXICOS EN LEVADURA 6 (ATL6) [41,42]. A comparative analysis of the effect of silencing and overexpressing *PvFER1*, *PvRALF1*, and *PvRALF6* on the number of nodules showed contrasting effects under different nitrate concentrations, indicating that the PvRALF1/6–PvFER1 complex may be positively or negatively involved in regulating the number of nodules, depending on the nutritional status of the plant (Figure 3 and Figure 6). Similar opposite functions have been previously documented for the RALFs and CrRLK1Ls; for instance, the RALF1–FER complex promotes root hair growth and inhibits root elongation [26,44]. Under biotic stress, the RALF17–FER complex acts as a positive regulator of plant defense responses, while RALF23/33/34–FER functions as a negative regulator [57]. In abiotic stress conditions, FER and the CrRLK1Ls HERCULES1, HERCULES2, THESEUS1, and MEDOS1–4 can act as positive or negative regulators of root development, but have an opposite function in the hypocotyls, in response to metal ions [58,59].

In the present study, we identified some differences in the nodule number and expression of genes related to the AON and NRN pathways when *PvFER1*, *PvRALF1*, and *PvRALF6* were differentially expressed. There was a direct association between changes in the expression of *PvCLE35*, *PvRIC2*, *PvNIC1*, *PvNARK*, or *PvTML* and the reduction in the number of nodules in silenced *PvFER1* and *PvRALF6* in 0 mM of nitrate, and overexpressing *PvRALF1* in roots grown with 5 mM of nitrate. Similarly, the effect of overexpressing *PvFER1* or *PvRALF1* in 0 mM of nitrate and silencing *PvFER1*, *PvRALF1*, or *PvRALF6* grown with 5 mM of nitrate had a consistent effect on increasing the number of nodules and changes in the expression of those genes from the AON and NRN pathways. In the remaining conditions (silencing of *PvRALF1* or overexpressing *PvRALF6* in 0 mM nitrate, and overexpression of *PvFER1* or *PvRALF6* at 5 mM nitrate), however, there is no obvious explanation for the observed phenotype based on the expression of the genes related to AON and NRN (Figure 4 and Figure 7). These observations strongly suggest the existence of additional participants or additional pathways involved in the regulation of the nodule number by PvFER1, PvRALF1, and PvRALF6 in *P. vulgaris*. This is not surprising as the versatility of FER functions in plant development has long been known [28,29,30]; for instance, the regulation of hormonal signaling by the RALF–FER complex may be an additional mechanism by which RALF and FER could regulate the number of nodules in *P. vulgaris*, since the importance of these signaling molecules has been reported in nodulation [60,61]. Another possibility could be based on the relationship between gene expression patterns and function. In this sense, by modifying the expression levels of *PvFER1*, *PvRALF1*, and *PvRALF6* in the roots, the normal expression patterns (the specific zone, level and/or condition of expression) of the genes of the AON and NRN pathways, could be altered. Therefore, it can be argued that beyond its function in nodules, there may be unexpected effects due to the changes in its expression patterns. For example, it is known that the function of gibberellins is specific as a function of time, and in the early stages of nodulation, functions as an inducer of nodule organogenesis, but its latter function is as an inhibitor of nodulation [61]. A similar effect may be happening due to changes in the genes evaluated here. Specific silencing or overexpression of *PvFER1*, *PvRALF1*, and *PvRALF6* will shed light on the mechanism by which these genes exert their function on nodulation through the AON and NRN pathways.

AON-related genes are also involved in the regulation of the number of nodules in *P. vulgaris* under low-phosphate conditions [62]. Furthermore, it was recently reported that RALF23 is induced by PHOSPHATE STARVATION RESPONSE 1 (PHR1) upon phosphate starvation in *A. thaliana* and, together with FER, participates in the suppression of plant immunity under low-phosphate conditions [56]. It would therefore be interesting to analyze whether *PvFER1*, *PvRALF1*, and *PvRALF6*, or other *CrRLK1L* and *RALF* genes in *P. vulgaris*, play a role in the regulation of nodule number as a function of phosphate concentration. The functional analysis of *FER* and *RALF* during nodulation under low-phosphate conditions, and in other different nutritional conditions, will shed light on the possible role of these ligand–receptor complexes as key regulators of the number of nodules required in legumes in relation to various nutritional environments.

In summary, in this work we demonstrated that *PvFER1*, *PvRALF1*, and *PvRALF6* are expressed during the nodulation process and presumably regulate the number of nodules in *P. vulgaris* as a function of nitrate levels. Based on these results, we propose a model that connects the functions in nodulation of *PvRALF1*, *PvRALF6*, and *PvFER1* in relation to nitrate availability (Figure 8), probably mediated by its modulation of the expression levels of CLE peptides and their NARK receptors in the roots.

## 4. Materials and Methods

### 4.1. Identification, Bioinformatic Analysis, and Phylogeny of P. vulgaris FER and RALFs

Following the same strategy used to identify CrRLK1L proteins in more than 50 plant species [34], RALF peptides from *A. thaliana*, *P. vulgaris*, *G. max*, and *M. truncatula* were identified using a BLASTP search of the Phytozome v12 database (https://phytozome.jgi.doe.gov, accessed on 10 October 2022) [63], using the RALF1 peptide sequence of *A. thaliana* as a query.

The peptide alignment of RALF amino acid sequences from the four aforementioned plant species was performed using the MUSCLE algorithm within the AliView alignment editor [64], followed by a manual optimization of misaligned regions. A maximum-likelihood phylogenetic tree [65] was constructed for the sequence alignment using IQ-TREE v1.6.12 [66] and the JTT+F+R10 substitution model, with 1000 bootstraps and default parameters.

The conservation of the protein motifs present in the four plants tested was determined using the MEME analyzer (http://meme-suite.org, accessed on 28 May 2020) [67]. This was carried out using the full-length amino acid sequences, setting the maximum number to 15 motifs, the number of motifs expected to any number of repeats, and the length of the motif as 10 to 200 amino acids. The other parameters were kept as default.

### 4.2. In Silico Expression Profile Analysis of RALF Genes in P. vulgaris

Expression profiles of the nine members of the *P. vulgaris RALF* gene family were retrieved from the *P. vulgaris* Gene Expression Atlas PvGEA (https://plantgrn.noble.org/PvGEA/, accessed on 5 March 2022) [40]. To better visualize these expression patterns, a heat map was made of the expression profiles of the nine genes. The distribution and abundance of differentially expressed genes were also presented as heatmaps using Heatmap Illustrator v1.0 (HemI) [68].

### 4.3. Plant Growth Conditions

*P. vulgaris* cv. Negro Jamapa seeds were surface-sterilized and incubated for 2 days at 28 °C in the dark. At 2 days post-germination (dpg), the seedlings were planted in pots of sterile vermiculite and inoculated with *R. tropici* CIAT 899 at an OD_600_ of 0.05 or with mock solution in the case of the control plants. The plants were irrigated with Fahraeus medium [69] with either 0 mM nitrate or 2 or 5 mM nitrate (nodule-inhibitory conditions) to analyze the expression, promoter activity, and function of the genes of interest. Roots at 3, 5, 7 and 14 days post-inoculation (dpi) were harvested, frozen in liquid nitrogen, and then stored at −75 °C until required for RNA extraction.

### 4.4. Composite Plant Generation

*P. vulgaris* seedlings (2 dpg) were inoculated with a selected clone of *Agrobacterium rhizogenes* carrying the plasmid construct of interest for promoter, silencing, or overexpression analysis of the gene of interest, as well as their respective controls. The inoculated seedlings were placed at the top of 15 mL Falcon tubes, with the cotyledon and the root outside and submerged into Fahraeus medium, respectively. Then, the seedling-containing tubes were introduced into 50 mL glass test tubes, filling with Fahraeus medium below the cotyledon to maintain the humidity. At 12 dpi with *A. rhizogenes*, transgenic roots expressing the fluorescent reporter protein were selected, and non-transformed and non-fluorescent transgenic roots were eliminated. Composite plants were transplanted into pots with vermiculite and inoculated with *R. tropici* CIAT899 (OD600 of 0.05) or mock-inoculated for further analysis.

### 4.5. RNA Extraction and qPCR Analysis

RNA was isolated from the frozen tissues using Trizol reagent, following the manufacturer’s instructions (MilliporeSigma, Burlington, MA, USA). The RNA integrity was verified using electrophoresis and its concentration was assessed using a NanoDrop2000 spectrophotometer (Thermo Fisher Scientific, Waltham, MA, USA). Any genomic DNA contamination was removed by incubating the samples with RNase-free DNase (10 U/µL; Roche, Basel, Switzerland) at 37 °C for 30 min. The cDNA was synthesized from 200 ng RNA using RevertAid Reverse Transcriptase (200 U/µL; Thermo Fisher Scientific). Next, a qPCR assay was performed using a Maxima SYBR Green/ROX qPCR kit (Thermo Fisher Scientific) on the Applied Biosystems QuantStudio 5 qPCR system (Thermo Fisher Scientific), following the manufacturer’s instructions. The thermal cycling conditions were as follows: 95 °C for 10 min, followed by 40 cycles of 95 °C for 15 s and 60 °C for 60 s. The melting curve stage was evaluated under the following thermal conditions: 95 °C for 15 s, 60 °C for 60 s, and 96 °C for 5 s. Relative expression values were calculated using the 2^−ΔCt^ method [70] with the widely used *elongation factor 1α* (*PvEF1α*) and the *P. vulgaris insulin-degrading enzyme* (*PvIDE*) as internal references [71,72]. Similar results were obtained using either *PvIDE* or *PvEF1α* as the reference gene. Only the *PvIDE* reference gene was selected to normalize the data in the figures. Three biological replicates were performed with three technical repeats. The gene-specific oligonucleotides used for the RT-qPCR assay are listed in Appendix A.

### 4.6. Plasmid Design and Construction

To analyze the activity of each promoter, at least 2000 bp of the promoter sequence upstream of the *PvFER1*, *PvRALF1*, or *PvRALF6* translation start site were amplified from *P. vulgaris* genomic DNA and then cloned into the pENTR/SD/D-TOPO vector (Thermo Fisher Scientific). A Gateway LR reaction was performed between the entry vector (pENTR-*pFER1*, pENTR-*pRALF1*, or pENTR-*pRALF6*) and the destination vector (pBGWSF7.0) [73] according to the manufacturer’s instructions (Thermo Fisher Scientific). The resulting expression vectors (pBG-*pFER*1, pBG-*pRALF1*, and pBG-*pRALF6*) were cloned into *A. rhizogenes* K599. The pBGWSF7.0 vector allows the expression of the *GUS* reporter under the control of the promoter under evaluation (*proPvFER1*::*GUS*, *proPvRALF1*::*GUS*, and *proPvRALF6*::*GUS*) to indirectly determine promoter activity through GUS enzymatic activity. An empty version of the pBGWSF7.0 vector was used as the control for promoter activity analysis, as previously reported [72,73].

For the gene overexpression analyses, the coding sequences (CDSs) of *PvFER1*, *PvRALF1*, and *PvFER6* were amplified from 2 dpg *P. vulgaris* root cDNA. The fragment containing each CDS was cloned into pENTR/SD/D-TOPO vectors (pENTR-FER1_CDS_, pENTR-RALF1_CDS_, or pENTR-RALF6_CDS_) and then recombined with the vector pH7FWG2D [73] using the Gateway system. The generated constructs (pH7-PvFER1_CDS_, pH7-PvRALF1_CDS_, and pH7-PvRALF6_CDS_) were introduced into *A. rhizogenes* K599. The pH7FWG2D vector is suitable for driving the expression of a chimeric protein fused with green fluorescent protein (GFP), under the control of the *35S* promoter. As a control vector, an empty version of the pH7FWG2D vector (identified as GUS in results), which expresses a GFP-GUS fusion was used for all the overexpression analyses [72,73].

For the RNAi constructs, an amplified 120-bp fragment of the *PvFER1*, *PvRALF1*, or *PvRALF6* 5′ untranslated region (UTR) was first cloned into the pENTR vector (pENTR-*FER1i*, pENTR-*RALF1i*, or pENTR-*RALF6i*) and then recombined with the ptdT-DC-RNAi vector (ptdT-*FER1i*, ptdT-*RALF1i*, or ptdT-*RALF6i*) [74]. ptdT-DC-RNAi allows the expression of a stem-loop RNA structure containing a sequence complementary to the target, which is processed by the RNA-induced silencing complex and then results in the silencing of the target. As the control vector, ptdT-DC-RNAi containing a scrambled DNA sequence (identified as SAC in results) was used for all the RNAi analyses, as reported in previous works [72,74,75].

All generated vectors were confirmed by sequencing. The sequences of the different oligonucleotides used are listed in Appendix A.

### 4.7. Analysis of the Promoter Activities

The promoter activities of the genes of interest were evaluated using the GUS staining protocol [76]. Roots containing one of the vectors for promoter analysis (pBG-*pFER1*, pBG-*pRALF1*, and pBG-*pRALF6*) were harvested at 7, 14 or 21 dpi. To examine the GUS activity, the roots were clarified, rehydrated, and mounted as described by Jefferson (1987). Treatment times were up to 1, 2, and 6 h for pBG-*pFER1*, pBG-*pRALF1*, and pBG-*pRALF6* composite plants, respectively. To rule out a false positive due to the GUS endophytic activity, control transgenic roots carrying the empty vector (EV) with no promoter to mediate GUS report expression, were also subjected to GUS staining for up to 24 h. No staining was observed, ruling out false positive.

The stained roots were observed using bright-field microscopy (02552240; AmScope, Irvine, CA, USA).

### 4.8. Analysis of the Number and Diameter of Nodules in Transgenic Roots

Composite plants expressing any of the different silencing, overexpressing, or control (*pTdT-SAC*, pH7FWG2D EV) constructs were inoculated with *R. tropici* CIAT899 GUS. These transgenic roots were harvested at 21 and 31 dpi under 0, 2, and 5 mM of nitrate conditions. The number and diameter of the nodules were analyzed using FIJI software [77,78].

### 4.9. Nitrogenase Activity Analysis

Nitrogenase activity was determined by measuring acetylene reduction [79,80]. Transgenic roots nodulated with *R. tropici* CIAT899 WT (21 dpi) were placed in 160 mL glass vials. After sealing the vials with rubber stoppers, 2 mL of air was withdrawn with a syringe and the same amount of acetylene was injected into each of the vials. The samples were incubated for 2 h at room temperature, and the ethylene production was measured in a gas chromatograph (Varian model 3300; Agilent Technologies, Santa Clara, CA, USA), as described by Ramírez et al. (1999) [80]. The nodules were then removed and dehydrated to measure their dry weight. Acetylene reduction was expressed as the µmol of ethylene h^−1^ g of nodule dry weight^−1^.

### 4.10. Protein Interaction Analysis

To examine whether the PvFER1 peptide interacts with PvRALF1 or PvRALF6, vectors were first constructed to perform the split ubiquitin system [81]. A modified version of the *PvFER1* CDS lacking the kinase domain was cloned into the pENTR/SD/D-TOPO vector (pENTR-FER1ΔK). The previously described pENTR-RALF1_CDS_ and pENTR-RALF6_CDS_ vectors were recombined with the destination vector MetYC_GW (Cub destination vector), while pENTR-FER1ΔK was combined with pXN22_GW (NubG destination vector) [82] for the split-ubiquitin system assays.

Yeast strains THY.AP4 (MATa ura3, leu2, lexA::LacZ::trp1 lexA::HIS3 lexA::ADE2) and THY.AP5 (MATα URA3, leu2, trp1, his3 loxP::ade2) were transformed with the PvFER1ΔK-Cub, PvRALF1-Nub, and PvRALF6-Nub constructs using the LiAc protocol previously described by Lalonde et al. (2010) [81]. Fusions that did not interact with soluble NubWT, which has a high affinity for the Cub domain, were considered false negatives, while those that interacted with NubΔG (which has a reduced affinity for Cub) corresponded to false positives [81].

### 4.11. Statistical Analysis

To establish the significance of the results obtained, differential statistical analyses were performed. Transcript accumulation data for all evaluated genes, the number of nodules, and the acetylene reduction data were analyzed using a non-parametric Mann–Whitney test. The diameters of the nodules were analyzed using a Kruskal–Wallis test, followed by Dunn’s multiple comparisons. All statistical tests were carried out using GraphPad Software version 8.2.263 (GraphPad Software, San Diego, CA, USA).

## Figures and Tables

**Figure 1 ijms-24-05230-f001:**
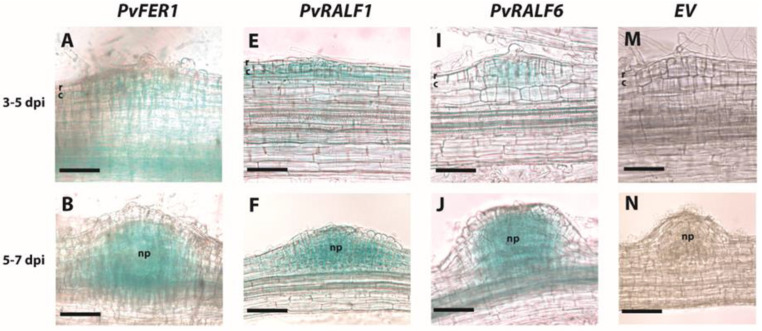
*PvFER1*, *PvRALF1*, and *PvRALF6* gene promoters are active during nodule organogenesis. Bright-field images of common bean transgenic roots expressing *proFER1*::GUS (**A**–**D**), *proRALF1*::GUS (**E**–**H**), or *proRALF6*::GUS (**I**–**L**). An empty vector (EV) was used as a control to rule out false positives (**M**–**P**). Promoter activity was visualized by GUS staining at different times after rhizobia inoculations (3–5, 5–7, 7–14, and 14–21 dpi). r, rhizodermis; c, cortex; np, nodule primordium; mn, mature nodule. The scale bar represents 500 μm.

**Figure 2 ijms-24-05230-f002:**
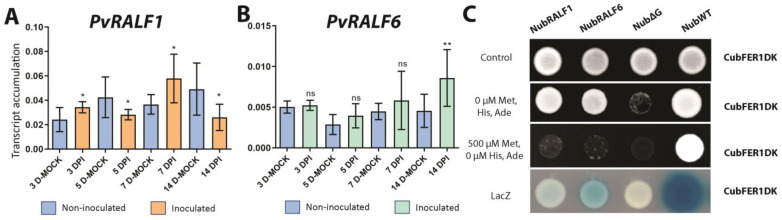
*PvRALF1* and *PvRALF6* respond to rhizobial inoculation at 0 mM and the corresponding peptide products interact with PvFER1ΔK. Accumulation levels of *PvRALF1* (**A**) and *PvRALF6* transcripts (**B**) in inoculated and non-inoculated wild-type roots under 0 mM nitrate. Data are means ± SD (*n* = 9). A non-parametric Mann–Whitney test was used to assess significant differences. * *p* < 0.05, ** *p* < 0.01; ns = not significant. The transcript accumulation of the *IDE* gene was used as a reference. (**C**) Split-ubiquitin system assays showing interaction between PvFERΔK and PvRALF1 or PvRALF6. NubWT and NubΔG were used as positive and negative controls, respectively. The experiments were repeated twice with similar results.

**Figure 3 ijms-24-05230-f003:**
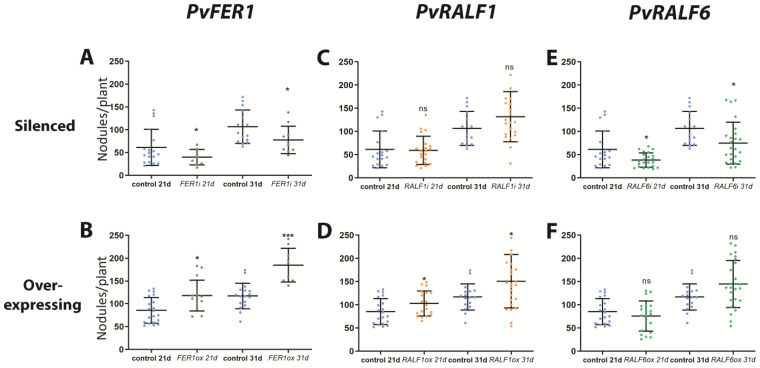
*PvFER1*, *PvRALF1*, and *PvRALF6* have a positive effect in regulating the nodule number, in 0 mM nitrate. (**A**–**F**) Quantification of the number of nodules in *PvFER1-* (gray dots, **A**), *PvRALF1-* (orange dots, **C**), and *PvRALF6*-silenced (green dots, **E**), and *PvFER1*- (gray dots, **B**), *PvRALF1*- (orange dots, **D**), and *PvRALF6*-overexpressing (green dots, **F**) transgenic roots at 21 and 31 dpi and under 0 mM nitrate, and their corresponding control roots (SAC for silenced and GUS for overexpressing roots, blue dots). Lines represent the mean ± SD. Data from three independent experiments are plotted as dots (*n* ≥ 9). A non-parametric Mann–Whitney test was used to assess significant differences. * *p* < 0.05, *** *p* < 0.001; ns = not significant.

**Figure 4 ijms-24-05230-f004:**
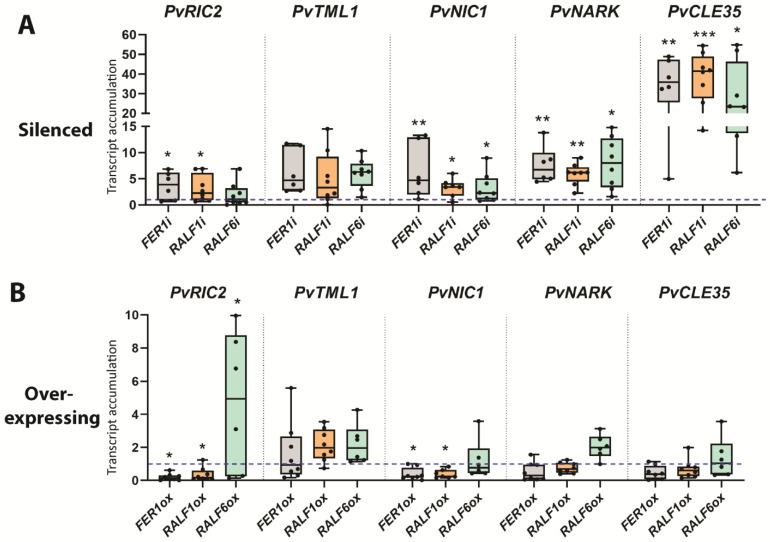
*PvFER1*, *PvRALF1*, and *PvRALF6* negatively regulate the expression of genes related to AON and NRN under low-nitrate conditions. (**A**,**B**) Transcript accumulation levels of AON (*PvRIC2* and *PvTML*) and NRN (*PvNIC*, *PvNARK*, and *PvCLE35*) marker genes in *PvFER1*- (gray boxes), *PvRALF1*- (orange boxes), and *PvRALF6*-silenced (green boxes) (**A**) or -overexpressing (**B**) transgenic roots at 21 dpi under low-nitrate conditions. The blue dotted lines represent the levels of the transcripts in the control roots for the five genes analyzed. The upper and lower edges of the boxes delimit the first to third quartiles, the horizontal line within the box represents the median, and the whiskers indicate the smallest and largest outlier in the data set. Data from three independent experiments are plotted as dots (*n* ≥ 8). Accumulation levels of *IDE* transcript were used as the reference. A non-parametric Mann–Whitney test was used to assess significant differences (* *p* < 0.05, ** *p* < 0.01, *** *p* < 0.001).

**Figure 5 ijms-24-05230-f005:**
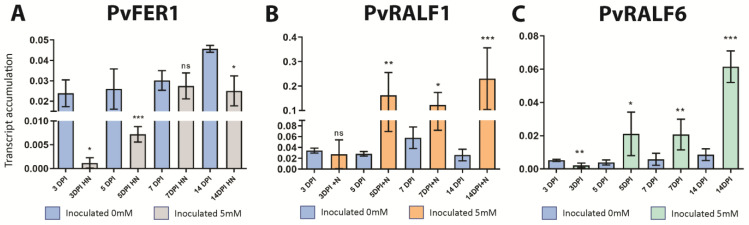
*PvFER1*, *PvRALF1*, and *PvRALF6* respond to rhizobial inoculation in roots at 5 mM. (**A**–**C**) Accumulation levels of *PvFER1* (gray bars, **A**), *PvRALF1* (orange bars, **B**), and *PvRALF6* (green bars, **C**) transcripts in roots inoculated in low (blue bars) and high nitrate. Data are means ± SD (*n* ≥ 9). A non-parametric Mann–Whitney test was used to assess significant differences. * *p* < 0.05, ** *p* < 0.01, *** *p* < 0.001; ns = not significant. The transcript accumulation of the *IDE* gene was used as a reference.

**Figure 6 ijms-24-05230-f006:**
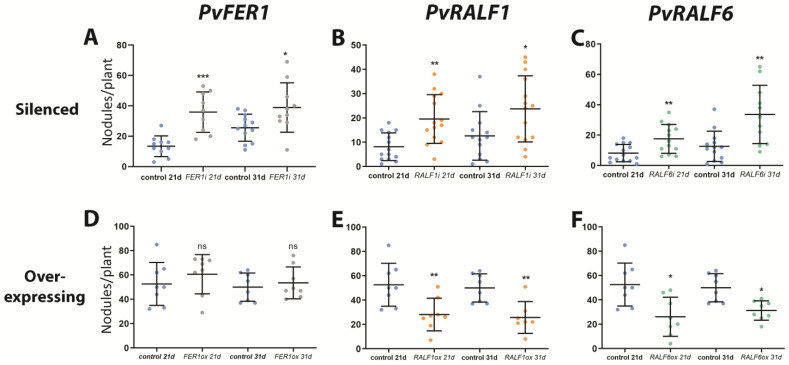
*PvFER1*, *PvRALF1*, and *PvRALF6* have a negative effect in regulating the nodule number in 5 mM nitrate. (**A**–**F**) Effect of silencing and overexpression of *PvFER1* (gray dots, **A**,**D**), *PvRALF1* (orange dots, **B**,**E**) and *PvRALF6* (green dots, **C**,**F**) on the number of nodules at 21 and 31 dpi. Blue dots are the controls in each case (SAC for silenced and GUS for overexpressing roots). The lines represent the mean ± SD. Data from three independent experiments are represented as dots (*n* ≥ 9). A non-parametric Mann–Whitney test was used to assess significant differences. * *p* < 0.05, ** *p* < 0.01, *** *p* < 0.001; ns = not significant.

**Figure 7 ijms-24-05230-f007:**
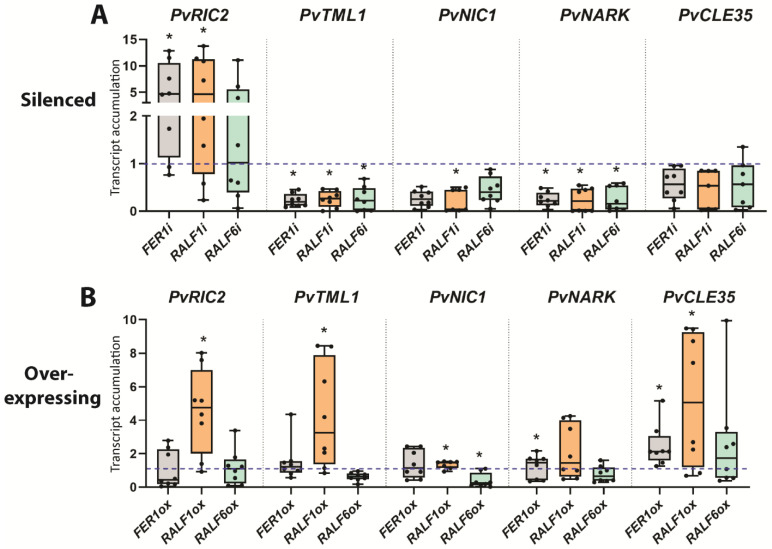
Positive regulation of genes related to AON and NRN by PvFER1, PvRALF1, and PvRALF6 under high-nitrate conditions. (**A**,**B**) Accumulation levels of AON (*PvRIC2* and *PvTML*) and NRN (*PvNIC*, *PvNARK*, and *PvCLE35*) marker gene transcripts in *PvFER1*- (gray boxes), *PvRALF1*- (orange boxes), and *PvRALF6*-silenced (green boxes) (**A**) and -overexpressed (**B**) transgenic roots at 21 dpi under 5 mM nitrate. The blue dotted line represents the levels of the transcripts in the control roots for the five genes analyzed. The upper and lower edges of the boxes delimit the first to third quartiles, the horizontal line within the box represents the median, and the whiskers indicate the smallest and largest outlier in the data set. Data from three independent experiments are represented as points (*n* ≥ 8). A non-parametric Mann–Whitney test was used to assess significant differences. * *p* < 0.05. The IDE transcript accumulation was used as the internal reference gene.

**Figure 8 ijms-24-05230-f008:**
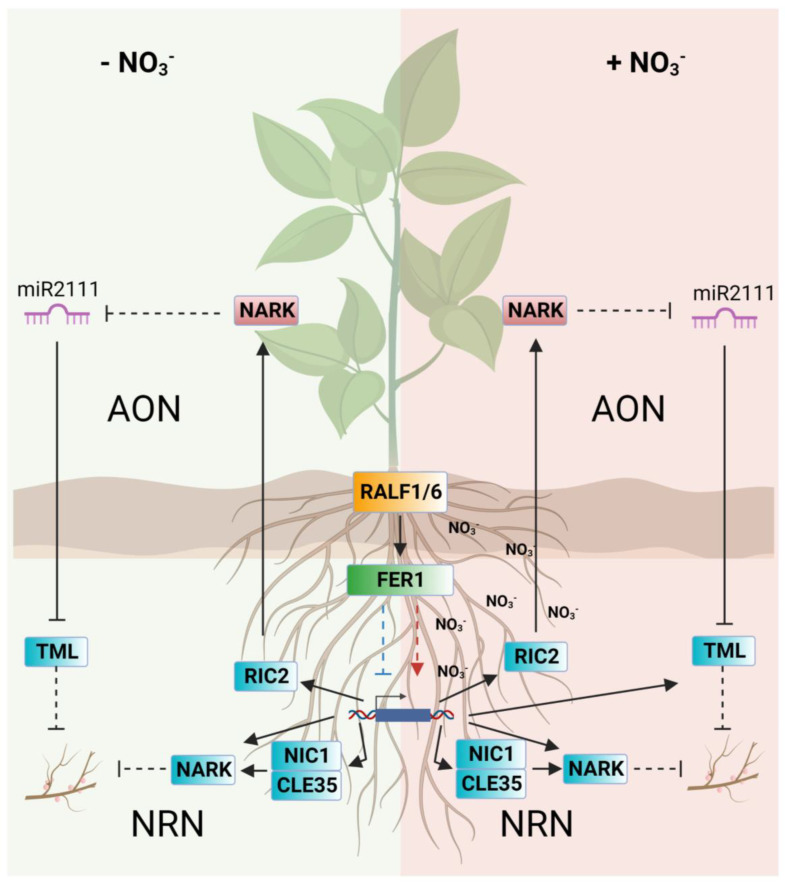
Proposed model for the regulation of the number of nodules in common bean by the PvRALF1/6-PvFER1 complex as a function of nitrate levels. In the absence of nitrate (left panel), RALF1 and RALF6 (orange box), and probably other RALFs, are perceived by FER1 (green box) and mediate the downregulation of the AON and NRN pathways, inducing the formation of more nodules. When nitrate is present (right panel), the RALF peptides are recognized by FER1 and mediate the upregulation of the AON and NRN pathways, leading to an inhibition of additional nodule formation. Solid lines indicate direct and dotted lines indicates indirect interaction. The arrowheads represent activation, and line endings indicate a repression. NO_2_^−^, nitrate; RALF1/6, RAPID ALKALINIZATION FACTOR 1/6; FER1, FERONIA 1; CLE35, CLAVATA3/ENDOSPERM-SURROUNDING REGION (CLE) 35; NIC1, NITRATE-INDUCED CLE 1; NARK, NODULE AUTOREGULATION RECEPTOR KINASE; RIC2, RHIZOBIA-INDUCED CLE 2; TML, TOO MUCH LOVE; AON, autoregulation of nodulation; NRN, nitrate-mediated regulation of nodulation.

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
