# Peer review of "The Phaseolus vulgaris Receptor-Like Kinase PvFER1 and the Small Peptides PvRALF1 and PvRALF6 Regulate Nodule Number as a Function of Nitrate Availability"

_ijms, 2023, doi:10.3390/ijms24065230_

Round 1

Reviewer 1 Report

The paper has uncovered a role for RALFs and FERONIA in promoting nodulation (RNAi of these genes reduced the number of nodules, while their overexpression increased nodule number) under low nitrate. The study suffers from lack stable mutants and use of RNAi. Nonetheless, the complementary use of overexpression, and use of two RALF peptides make the data convincing. In addition, an interesting negative correlation with RALF/FER and CLE/TML in the AON pathway is made. Intriguingly, RALF/FER seem to promote nodulation at high nitrate, although it is not entirely clear how this works.

The author’s explanation for the differential effect of RALF overexpression in Medicago vs Phaseolus seems reasonable, since their data suggests that infection is affected, and infection is persistent in Medicago’s indeterminate nodules and transient in Phasoleus’ determinate nodules. The discussion is thoughtful and interesting, in particular the comparisons with Arabidopsis, which suggests the link between RALF/FER and CLE/TML is conserved.

One short-coming in the analysis is that NARK/SUNN/HAR1 expression in the root is irrelevant to nodulation, it affects nodulation via the shoot. Ideally, shoot expression would be provided or the data should be removed.

The discussion would benefit from some additional consideration of where the genes are expressed (nodules vs root), and whether their expression at high and low nitrate is comparable. For instance, at high nitrate NICs are suppressed, so small changes in their expression are unlikely to be relevant, whilst at low nitrate their expression is relatively high, and they are major players. Genes that are mainly expressed in nodules (RICs) may have lower expression at 5mM nitrate since there are fewer nodules etc.

In Fig 8, Doesn’t Fer-ox induce expression of RIC2 at high N? This should be added, even if it can’t be explained.

In Fig 4 and 7 (A and B) the genes should be provided in the same order to make them easier to compare.

Figure 7 seems to be labeled wrong: A on the left side is labeled as silenced (i), but the X-axes are all labeled overexpression (ox), and B has the same problem. The text says A is the silencing data, so  presumably the X-axes labels in A and B are wrong.

In Fig 6a,b, c the ‘i’ seems missing Fer1i, Ralf1i etc. in the x-axis labels

The sentences that begin on lines 21 and 23 of the abstract are completely redundant, I suggest deleting the first since the second is more informative.

Within the text the figures have been called out of order (i.e .Fig 2A,B called before Fig. 1E-L)

Fig 3 would be easier to read if both controls were labeled simply as controls rather than as SAC and GUS, the nature of the controls can be explained in the legend.

Line 186 “using the split ubiquitin”

Line 187/ Cub-PvFER1ΔK needs to be briefly explained (it is only explained later in the M&M)

Line 609. “..were also subjected to GUS staining…”

Line 424. The sentence is a bit cumbersome as written, and I had to read it twice to sort out what was meant. Perhaps you could just say that numerous PvNARK-like LRR receptors were differentially expressed, since they can get the details in the supplemental figure anyway.

Line 450, 451 I think HERCULES is mis-spelled twice

Line 461 Overexpressing PvRALF6 in 0 mM nitrate was mentioned as ‘one of the remaining conditions’, but was already explained two sentences above.

In lines 457 and 461 rather than saying ‘the effect on the number of nodules’ and ‘consistent effect’ please be more precise and say what the effect is.

Reviewer 2 Report

The manuscript “ijms-2178213” entitled “The Phaseolus vulgaris receptor-like kinase PvFER1 and the small peptides PvRALF1 and PvRALF6 regulate nodule number as a function of nitrate availability)” by Solís-Miranda et al. provides an interesting study where it was found that the receptor-like kinase PvFER1 and the small peptides PvRALF1 and PvRALF6 participate in the regulation of the number of nodules in P. vulgaris as a function of nitrate availability. Functional analysis revealed that PvFER1, PvRALF1, and PvRALF6 act as positive regulators of nodule number in growth medium containing 0 mM nitrate but as negative regulators in medium with 2- and 5-mM nitrate. In addition, the effect on nodule number was found to be consistent with changes in the expression levels of genes associated with the AON pathway and with the nitrate-mediated regulation of nodulation (NRN).

For publication in the “International Journal of Molecular Science”, the topic and content are appropriate. The subject of the study is interesting and topical, with high scientific and practical importance. The introduction is in accordance with the subject and correctly presented. Numerous scientific articles of recent date and in concordance with the topic of the study were consulted. The methodology of the study was clearly presented, and appropriate to the proposed objectives. The obtained results have been fully analyzed. The scientific literature, to which the reporting was made, is recent and representative in the field. The editing and linguistic quality are good. In addition, it is easy to follow by the reader, the figures and tables give good summaries and the text editing to a thoughtful conclusion part. However, there are some points that need attention in order for the article to be published. I would like to recommend the publication of this article, and a (very) minor revision is required for the reasons listed below:

·      Figures 2c and 8: The quality of the figures is low and blurred. Please fix this problem.

·      Lines 649-650: Please provide information about the manufacturer of statistical software as in other research materials of your study.

·      Be consistent with the formatting of references and cross-references. This has to be standardized across the paper. 

·      Finally, the reviewer recommends the authors carefully check and revise the references section. Specifically, delete lines 688-699 since these sentences are instructions (examples) for the references section (MDPI template).

Thank you for your consideration.
